



# Basal buoyancy and fast moving glaciers: in defense of
# analytic force balance
C.J. van der Veen
Department of Geography and Atmospheric Science, University of Kansas, 203 Lindley Hall, 1475
Jayhawk Blvd, Lawrence, KS 66045
**Abstract.** The geometric approach to force balance advocated by T. Hughes in a series of publications has
challenged the analytic approach by implying that the latter does not adequately account for basal buoyancy
on ice streams, thereby neglecting the contribution to the gravitational driving force associated with this
basal buoyancy. Application of the geometric approach to Byrd Glacier, Antarctica, yields physically
unrealistic results and it is argued that this is because of a key limiting assumption in the geometric
approach. A more traditional analytical treatment of force balance shows that basal buoyancy does not
affect the balance of forces on ice streams, except locally perhaps, through bridging effects.





**1. Introduction**
Ice streams are fast-moving rivers of ice embedded in the more sluggish-moving main body of ice sheets,
and are responsible for the bulk of drainage from the interior in West Antarctica.  Most ice streams start
well upstream from the coast, some extending several hundreds of km into the interior, and drain into
floating ice shelves or ice tongues and are believed to represent the transition from inland-style "sheet
flow" to ice-shelf spreading.  The nature of this transition remains under debate, however.
In a long series of papers, T. Hughes presents the geometric approach to the balance of forces acting on ice
shelves, ice streams, and interior ice [*Hughes*, 1986, 1992, 1998, 2003, 2009a, 2009b, 2012*; Hughes et al.*,
2011, 2016].    Rather than working his way through the basic equations, as done by most other
investigators, including *Van der Veen and Whillans* [1989] and *Van der Veen* [2013], he presents
derivations based on graphical interpretation of triangles representing forces acting on an ice column.  In
essence, the transition in flow regime is achieved by introducing a basal buoyancy factor that describes the
gradual ice-bed decoupling towards the grounding line.
The idea of basal buoyancy has been invoked many times before in glaciology, in particular in the context
of formulating a sliding relation.  In many models, the sliding speed is assumed to be inversely proportional
to the "effective basal pressure" defined as the difference between the weight of the overlying ice and the
pressure in the subglacial drainage system.  Intuitively, this approach may seem to make sense: as the
subglacial water pressure increases, the normal force on the bed should be reduced, thus allowing the
glacier to move faster.  An analogy may be drawn with pushing a shopping cart across a sandy beach: the
less groceries are in the cart, the easier it is to push the cart forward.  The difference is, of course, that the
weight of the groceries is pre-determined (by the ice thickness), so the only way to facilitate the forward
motion is through some force acting to lift the cart upward.  *Hughes* [2008, 2012] suggests that basal
decoupling provides this upward force.
The objective of this brief note is to evaluate the implications of Hughes' geometric approach to force
balance by applying the results to Byrd Glacier, East Antarctica.
**2. Force balance: analytical approach**
Analytical treatments of glacier force balance are numerous and derivations of the depth-integrated force-
balance equations are now standard fare in most glaciology textbooks.  In most cases, this balance of forces
is discussed in terms of stress deviators, defined as the full stress minus the hydrostatic pressure.  This is
done because the flow law for glacier ice relates strain rates to stress deviators.  That is
$$\sigma'_{ij} = \sigma_{ij} - \frac{1}{3}\delta_{ij}\left[\sigma_{xx} + \sigma_{yy} + \sigma_{xx}\right] \tag{1}$$



where the prime denotes the stress deviator and unprimed stresses are full stresses, and $\delta_{ij} = 1$ for i = j and
$\delta_{ij} = 0$ for i ≠ j. Deviatoric stresses are called for in the flow law for glacier ice because the rate of
deformation is in good approximation independent of the hydrostatic pressure. However, the use of
deviatoric stresses in discussing the balance of forces unnecessarily complicates the interpretation because
the longitudinal deviatoric stress in one direction depends on the full normal stresses in all three directions
of a Cartesian coordinate system. It is more convenient to consider stresses in a glacier as the sum of the
stress due to the weight of the ice (lithostatic stress) and stresses, $R_{ij}$, due to the flow (resistive stresses).
This partitioning makes a clearer distinction between action and reaction in glacier dynamics [*Whillans*,
1987] and follows common practice in geophysics [*Engelder*, 1993, p. 10; *Turcotte and Schubert*, 2002, p.
77].
It may be noted that the term "resistive" stress is an unfortunate choice, perhaps, because these stresses do
not necessarily always offer resistance to flow. For example, gradients in longitudinal stress can act in
cooperation with the driving stress in pulling the ice forward. A more appropriate terminology would
perhaps be *flow stress* or, following geophysical terminology, *tectonic stress.* The $R_{ij}$ represent the
stresses that are associated with glacier deformation, as opposed to the lithostatic stress which describes the
action of gravity. However, the existing terminology appears to have made its way into the glaciological
literature [e.g. *Cuffey and Paterson,* 2010 section 8.2.2] and a name change at this stage likely would
introduce even more confusion.
*Van der Veen* [2013, sect. 3.1] presents a derivation of the column-average balance equations by integrating
the momentum balance equations over the full ice thickness. *Van der Veen and Payne* [2004] and *Van der*
Veen [2013, sect. 3.2] present a discussion of force balance based on geometric arguments and, not
surprisingly, arrive at the same result. Without loss of generality, flow in one horizontal direction may be
considered. That is, the horizontal x-axis is chosen in the direction of flow and it is assumed that there is
no component of flow in the other horizontal y-direction. The z-axis is vertical upward, with z = 0 at sea
level. Force balance in the flow direction is then described by the following equation [*Van der Veen and*
*Whillans*, 1989; *Van der Veen*, 2013, sect. 3.1]:
$$\tau_{dx} = \tau_{bx} - \frac{\partial}{\partial x}\left(H\tilde{R}_{xx}\right) - \frac{\partial}{\partial y}\left(H\tilde{R}_{xy}\right) \tag{2}$$
In this expression, $\tau_{dx}$ denotes the gravitational driving stress, defined as
$$\tau_{dx} = -\rho g H \frac{\partial h}{\partial x} \tag{3}$$
where ρ represents the density of ice, g the gravitational acceleration, H the ice thickness, and h the
elevation of the upper ice surface. The terms on the right-hand side of equation (2) represent the resistance
to flow associated with, respectively, drag at the glacier base, gradients in longitudinal stress ("pulling
power") and lateral drag arising from shear between the faster-moving ice stream and the near-stagnant



interstream ridges or fjord walls. The tilde (~) denotes depth-averaged values. Resistive stresses are
defined following *Van der Veen and Whillans* [1989] as:
$$R_{xx} = \sigma_{xx} + \rho g(h - z) \tag{4}$$
$$R_{xy} = \sigma_{xy} \tag{5}$$
where $\sigma_{ij}$ represents the full stress, and $-\rho g(h - z)$ the lithostatic stress (weight of the ice above) at depth z.
The balance equation (2) is exact. No approximations are involved in deriving this expression from the
basic equations describing the balance of forces on a segment of ice [*Van der Veen and Whillans*, 1989;
*Van der Veen*, 2013, sect. 3.1]. Consequently, this equation applies to free-floating ice shelves where the
gravitational driving stress is balanced entirely by gradients in longitudinal stress, yielding the classic
*Weertman* [1957] solution [*Van der Veen*, 2013, sect. 4.5], as well as laminar flow with basal drag
providing sole resistance to flow [*Van der Veen*, 2013, sect. 4.2]. Except for these two end-member
solutions, equation (2) does not permit analytical solutions without making additional assumptions.
Nevertheless, because no approximations were made in its derivation, balance equation (2) applies equally
well to transitory flow regimes such as ice streams and outlet glaciers.
Integrating the balance equation over the width of the flowband simplifies the resistive term associated with
drag at the lateral margins. Denoting the lateral shear stress at the margins by $\tau_s$ (assumed to have the
same magnitude but opposite signs at both lateral margins), and glacier width by W, lateral resistance on a
section of glacier of unit width is [*Van der Veen*, 2013, eq. (4.39)]
$$F_s = \frac{2H\tau_s}{W} \tag{6}$$
and the width-averaged force-balance equation becomes
$$\tau_{dx} = \tau_{bx} - \frac{\partial}{\partial x}\left(H\tilde{R}_{xx}\right) + \frac{2H\tau_s}{W} \tag{7}$$
with the understanding that all terms are averaged over the flowband width (or, equivalently, considered
constant across the flowband). Note that contrary to what *Hughes* [2008, p. 53] states, lateral drag does not
vanish at the center of a glacier. While the shear stress, $R_{xy}$, is zero at the centerline, its transverse
derivative and thereby resistance from lateral drag, is not zero there. In fact, according to equation (6), this
resistance is constant across the glacier width.
The geometric approach developed by Hughes arrives at a similar balance equation, namely
$$-\rho g H \frac{\Delta h}{\Delta x} = \tau_b - \frac{\Delta H \sigma_F}{\Delta x} + \frac{2H\tau_s}{W} \tag{8}$$
[*Hughes*, 2003, eq. (36)] or, taking the limit $\Delta x \rightarrow 0$





$$-\rho g H \frac{\partial h}{\partial x} = \tau_b - \frac{\partial H \sigma_F}{\partial x} + \frac{2 H \tau_s}{W} \qquad (9)$$
In these balance equations, $\sigma_F$ is related to the deviatoric tensile stress; its exact interpretation has evolved
over the years. To avoid unnecessary confusion, a consistent notation is used in the following discussion,
based on *Hughes* [2008, 2012]. Comparison of equations (7) and (9) shows that $\sigma_F = \tilde{R}_{xx}$. It is the way
this stress is calculated that sets Hughes' geometric approach apart from the analytical approach. In
essence, this stress is linked to basal buoyancy and, in later versions, downglacier-integrated resistance
from basal and lateral drag. While the force balance equation (7) does not imply any assumption about the
depth-variation in the longitudinal resistive stress, $R_{xx}$, *Hughes* [2003] explicitly argues that both $\sigma_F$ and
the associated stretching rate, $\dot{\varepsilon}_{xx}$, must be constant in the vertical direction.
**3. Force balance: geometric approach**
Discussing force balance for stream flow, *Hughes* [2008, section 11] equates $\sigma_F$ with a basal buoyancy
factor, $\phi$, as
$$\sigma_F = \frac{\rho g H}{2} \phi^2 \qquad (10)$$
where
$$\phi = \frac{\rho_w H_w}{\rho H} = \frac{P_w}{P_i} \qquad (11)$$
is determined by the ratio of the areal average water pressure under the ice, and basal ice pressure (or
weight of the ice column); $\rho_w$ represents the density of sea water. For a floating ice shelf, $\phi = 1$, and
expression (10) reduces to the solution for a free-floating ice shelf spreading in the x-direction only
[*Weertman*, 1957; *Van der Veen*, 2013, sect. 4.5]. For inland-style flow, $\phi = 0$, and the lamellar flow
solution can be derived. For ice streams and outlet glaciers that represent the transition from interior-style
flow to ice-shelf spreading, $1 < \phi < 0$. In first-order approximation
$$\phi = \frac{H_o}{H(x)} \qquad (12)$$
where $H_o$ represents the thickness at the grounding line, and $H(x)$ the ice thickness at some distance x
upstream of the grounding line [*Hughes*, 2008, eq. (11.11)]. This relation is robust and a decrease in $\phi$
going upglacier from the grounding line increases ice-bed coupling and generally yields a concave surface
profile [*Hughes*, 2008, p. 58].
*Hughes* [2008] takes the geometric approach to another level and relates *all* resistance to flow on ice
streams to the basal buoyancy factor, $\phi$. In addition to relating the longitudinal stress deviator to this factor,





lateral and basal drags are linked to $\phi$ as [*Hughes*, 2008, table 12.1; see also *Hughes*, 2009a,b; *Hughes*,
2012, table 12.1; *Hughes et al.*, 2016, eqs. (12) – (17)]

$$\tau_b = -\rho g H (1-\phi)^2 \frac{\partial h}{\partial x} - \rho g H^2 (1-\phi)\frac{\partial \phi}{\partial x} \qquad (13)$$

$$F_s = -2\rho g H \phi (1-\phi)\frac{\partial h}{\partial x} - \frac{1}{2}\rho g H W (1-2\phi)\frac{\partial \phi}{\partial x} \qquad (14)$$

while the longitudinal stress gradient term is given by

$$\frac{\partial H \sigma_F}{\partial x} = \rho g H \phi \left( \phi \frac{\partial h}{\partial x} + H \frac{\partial \phi}{\partial x} \right) \qquad (15)$$

The achievement here is that these equations are derived without consideration of ice velocity or physical
properties of the ice (temperature, stiffness, fabric development, etc.), or, for that matter, basal water
availability and balance. Presumably, all these factors are somehow reflected in the ice-stream geometry
and the inferred basal buoyancy.
**4. Geometric approach: application to Byrd Glacier, Antarctica**
Balance of forces on Byrd Glacier, East Antarctica, was first discussed by *Whillans et al.* [1989] who used
measurements of surface velocity and surface topography derived from repeat aerial photogrammetry, to
evaluate the relative roles of lateral drag, gradients in longitudinal stress, and basal drag in resisting the
gravitational driving stress. *Van der Veen et al.* [2014] reconsidered these calculations and also
investigated the effect of drainage of two sub-glacial lakes in the catchment region. Both studies employed
the analytical force-balance approach.
*Reusch and Hughes* [2003], *Hughes* [2009a], *Hughes et al.* [2011], and *Hughes et al.* [2016] discuss force
balance on Byrd Glacier from the geometrical perspective and take issue with the analytical approach of
*Whillans et al.* [1989]. None of these studies explicitly shows how the various resistive forces vary along
the glacier and, instead, largely base their discussion on how the basal buoyancy, $\phi$, varies upstream of the
grounding line. Therefore, to fully appreciate the implications of the geometrical approach, equations (13)
– (15) are applied here to evaluate all terms in the balance of forces.
The geometry is shown in Figure 1 [*Van der Veen et al.*, 2014, fig. 6]. Only the lower 30 km stretch
upstream of the grounding line (at x = -10 km) is considered here because that is the region laterally
bounded by near-parallel ford walls. Also shown in Figure 1 is the basal buoyancy factor calculated from
eq. (12); $\phi$ increases from around 0.7 a little more than 30 km upstream of the grounding line, to 1 where
the ice starts to float. While there is nothing in particular wrong or disturbing about this basal buoyancy
factor, the situation becomes more problematic when the actual forces are considered.
The average driving stress is ~160 kPa, but shows large spatial variations that appear to be temporally fixed
(Figure 2). Gradients in longitudinal stress are mostly negative, averaging -140 kPa along the flowline,



implying that, except in a few isolated locations, this term acts in the same directions as the driving stress,
draining the grounded ice into the Ross Ice Shelf. To maintain balance of forces, flow resistance is
partitioned between basal drag (~53 kPa) and lateral drag (~247 kPa). In the geometric approach, the bulk
of flow resistance is associated with lateral drag and basal drag supports only about 1/3 of the driving
stress. This result is surprising and there is no credible physical mechanism that can explain this. Even on
a free-floating ice shelf, where other sources of flow resistance may be neglected, gradients in longitudinal
stress arising from water pressure act to oppose the driving stress [*Weertman,* 1957; *Van der Veen*, 2013,
sect. 4.5]. *Hughes et al.* [2016, p. 201] argue that the water buttressing produces a backstress in the
longitudinal force balance, and that this is a real stress that is obscured using continuum mechanics in the
conventional analytical approach. According to *Hughes* [2008, 2012], this stress, or "pulling power"
results in the overestimation of longitudinal stress gradients, adding to the driving stress.
**5. Limitation of the geometric approach**
To understand the limitation in the geometrical approach to force balance, consider the forces along an ice
stream flow line as discussed in *Hughes* [2008, p. 53 ff.] (see also figure 1 in *Hughes* [2003], and *Hughes*
[2012, section 11]). The geometry is shown in Figure 3. While *Hughes* [2008, p. 53; 2012, p. 66]
erroneously states that resistance from lateral drag vanishes at the centerline of an ice stream and therefore
does not include this source of resistance in his discussion, this has no significant impact on the following
discussion – lateral drag can be readily added to the basal drag term without altering the general tenets of
the analysis.
According to *Hughes* [2008, 2012], the gravitational driving force at x is

$$F_g = \text{area ADF} = \frac{1}{2}\rho g H^2 \qquad (16)$$

and this force must be balanced by longitudinal resisting forces consisting of a "water buttressing force"
(area CDE), a tensile force (area BCE), and a basal drag force (area ABEF). The basal drag force equals
integrated basal resistance from the grounding line to the upglacier location (integrated resistance from
lateral drag could also be included in this term). The area of each triangle is obtained from the familiar
formula (base × height) / 2, where the base either equals the ice overburden pressure (DF = ρgH) or water
pressure (DE = $\rho_w gH$), and the height equals the ice thickness (AD = H), flotation height (BD =
$H_f = (\rho_w/\rho)H_w$), or the piezometric height (CD = $H_w = P_w/(\rho_w g)$). Thus, each of the resistive
terms can be evaluated as a function of local ice thickness and water pressure. The reason why, for
example, area ABEF should be associated with basal drag force (or basal plus lateral drag), remains unclear
but is irrelevant.
The problem with this reasoning is that $F_g$ *does not* represent the gravitational driving force. Rather, this
force equals the lithostatic force associated with the weight of ice. When considering horizontal forces at
any location, this force is balanced exactly by an equal but opposite force from ice of equal thickness on the



left of the vertical line AD, except at the calving front. In other words, adhering to the geometric
representation, triangle ADF is balanced by the mirror triangle ADP (Figure 4a), whether one considers an
ice shelf, ice stream, or interior ice. The gravitational force that drives glacier flow is associated with
*gradients* in lithostatic stress (Figure 4b). A correct geometry-based discussion of force balance would
consider the difference between lithostatic stress at x and at some location x + Δx downglacier, and, in the
case of a sloping bed, lithostatic stress acting on the bed, and the difference between longitudinal stress at
both locations, in addition to basal and lateral drag acting over the distance considered. Doing so gives the
balance equation (7) [*Van der Veen and Payne*, 2004; *Van der Veen*, 2013, section 3.2].
It is *not* possible to relate resistive forces at any location to *point* values such as basal water pressure or
weight of the ice at location x. While resistive stresses, such as $R_{xx}$, can be evaluated at specific points,
resistance to flow is associated with *gradients* in these stresses [see, e.g., *Van der Veen*, 2013, figure 3.1
and eqs. (3.8) – (3.9)]. Balance of forces is only meaningful if applied to flowline segments, not single
locations. Consequently, the concept of force balance at any location is inherently flawed. While many, if
not most, glaciologists, *Van der Veen* [2013] included, often refer to driving stress or basal drag at location
x, it would be more appropriate to refer to these quantities as areal averages. If the surface slope is
calculated over a distance 2Δx, the associated driving stress is the average over the interval (x – Δx, x +
Δx), and similarly for basal drag. Nuancing common parlance to reflect this subtlety would render many
discussions of glacier dynamics unnecessarily cumbersome and should be superfluous for most readers
understanding the fundamentals of glacier dynamics.
**6. Discussion**
While the geometric force balance approach is severely limited, it is worth exploring the central premise of
Hughes' ideas, namely that the transition from sheet flow to shelf flow is achieved through basal buoyancy,
with interior ice firmly grounded on bedrock and ice shelves floating in sea water. It should be noted that
for both these end member solutions, at any location the weight of an ice column is fully supported from
directly below: terra firma in the case of grounded ice, and sea water for ice shelves.
While not immediately obvious, the role of varying subglacial water pressure is included in the force-
balance equation (7), namely though bridging effects [*Van der Veen*, 2013, sect. 3.4]. To clarify this,
consider that resistive stresses are linked to strain rates, or velocity gradients, by invoking Glen's flow law
for glacier ice [*Van der Veen and Whillans*, 1989; *Van der Veen*, 2013, sect. 3.3]:
$$R_{xx} = B\dot{\varepsilon}_e^{1/n-1}\left(2\dot{\varepsilon}_{xx} + \dot{\varepsilon}_{yy}\right) + R_{zz} \tag{17}$$

$$R_{xy} = B\dot{\varepsilon}_e^{1/n-1}\dot{\varepsilon}_{xy} \tag{18}$$



Here, B represents the temperature-dependent rate factor, and n = 3 the flow-law exponent; $\dot{\varepsilon}_e$ is the
effective strain rate defined as the second invariant of the strain-rate tensor. The last term on the right-hand
side of equation (17) is the vertical resistive stress defined as
$$R_{zz}(z) = \sigma_{zz} + \rho\,g\,(h - z) \tag{19}$$

For brevity of notation, the along-flow resistive stress is written as the sum of a contribution associated
with along-flow gradients in velocity (first term on the right-hand side of equation (17)) and the vertical
resistive stress:
$$R_{xx} = R_{xx}^{(0)} + R_{zz} \tag{20}$$

Force-balance in the horizontal direction can then also be written as
$$\tau_{dx} = \tau_{bx} - \frac{\partial}{\partial x}\left(H\tilde{R}_{xx}^{(0)}\right) - \frac{\partial}{\partial y}\left(H\tilde{R}_{xy}\right) - \frac{\partial}{\partial x}\int_{h-H}^{h} R_{zz}(z)\,dz \tag{21}$$

Where the weight of the ice is fully supported by the substrate below, the vertical resistive stress is zero.
This is the assumption usually made when considering the budget of forces acting on glaciers [e.g. *Van der*
*Veen and Whillans*, 1989]. Locally, however, bridging effects may be important, for example where a
water-filled cavity exists at the ice-bed interface [*Van der Veen*, 2013, sect. 7.2]. Where cavitation occurs
and basal ice becomes separated from the bed, the cavity cannot support the weight of the ice leading to
shear-stress gradients that effectively transfer the weight to surrounding areas where the ice is in contact
with the bed, such that the areal average of the vertical resistive stress is zero. Thus, on a large scale, such
as the length of ice streams and outlet glaciers, basal buoyancy is a non-issue where horizontal force
balance is concerned. Indeed, *Hughes* [1998, eq. (3.5)] does not include bridging effects in his discussions
and equates the total vertical stress at depth to the lithostatic stress.
Basal buoyancy may be important on ice streams and outlet glaciers according to the commonly-adopted
sliding relation in which sliding speed is inversely proportional to the effective basal pressure. *Pfeffer*
[2007] suggests that this proportionality may explain rapid velocity increases on tidewater glaciers and
Greenland outlet glaciers: as these glaciers thinned and thickness approached flotation, the effective basal
pressure approached zero, resulting in a large increase in sliding velocity. Another possibility is that
increased basal buoyancy reduces basal drag, thereby allowing glaciers to move faster. The importance of
these effects can be evaluated from analysis of time series of surface speed and glacier geometry, or using
numerical models based on the balance equation (7).
The primary difference between shelf flow and stream flow is not that on ice shelves the ice weight is
supported by water and on grounded interior ice this weight is supported by the bed below. The main
difference is that, because ice shelves float in water, basal drag is zero and resistance to flow must be
partitioned between gradients in longitudinal stress and lateral drag, whereas for sheet flow, basal drag
provides most resistance to flow. Thus, it would seem reasonable to propose that the transition from sheet
to shelf flow involves a gradual reduction in basal resistance, perhaps associated with the presence within



deforming sediments, or gradual drowning of bed obstacles. As basal drag becomes less important,
longitudinal stress gradients and lateral drag must increase and provide most or all resistance to the flow of
ice streams.
**7. Concluding remarks**
The geometrical approach to ice sheet modeling links ice-bed coupling directly to the stresses that resist
horizontal gravitational motion [*Hughes*, 2008, p. 34]. This basal buoyancy supposedly translates into a
major component of gravitational forcing by which ice sheets discharge ice into the sea [*Hughes*, 2003].
The concept as presented by Hughes in a series of publications spanning the last 30 years has yet to come
up with a solution that can be successfully applied to ice streams and outlet glaciers. This is not to say that
a geometric approach is inherently flawed – if implemented correctly it should produce consistent and
correct results but this has yet to be achieved.
The charge that the analytical force-budget approach fails to account for basal buoyancy and excludes a
"water buttressing force" on ice streams is incorrect. Equation (7) describing the depth-integrated balance
of horizontal forces is derived without making any simplifying assumptions and applies equally well to
floating ice shelves and firmly grounded interior ice. If some phantom force is missing from this equation,
this force must also be missing from the momentum balance equations that form the starting point for
deriving equation (7).
Hughes is correct that ice streams and outlet glaciers represent the transition from sheet flow and shelf flow
and that much remains to be understood about the nature of this transition. Advantageously, ongoing rapid
changes on many of the outlet glaciers have been well documented through time series of surface elevation
and surface velocity. The latter, in particular, are powerful indicators of the distribution of stresses on
glaciers because strain rates (velocity gradients) are directly linked to stresses through the flow law for
glacier ice. Improved understanding of the dynamics of rapidly-changing ice-sheet components will come
from interpretation of strain rates and temporal changes therein.
**8. Acknowledgements**
This note originated in 2005 in response to discussions with T. Hughes. An early version was submitted in
2009 to the *Journal of Geophysical Research* but was summarily rejected for consideration because the
Editor deemed it a personal attack on T. Hughes. The latest round of papers by T. Hughes prompted the
resurrection of this manuscript. I am indebted to Ken Jezek for his continued support and careful reading,
and to Leigh Stearns for additional comments. This research was supported NASA grant no.
FED0066542 and UNI0072622.





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



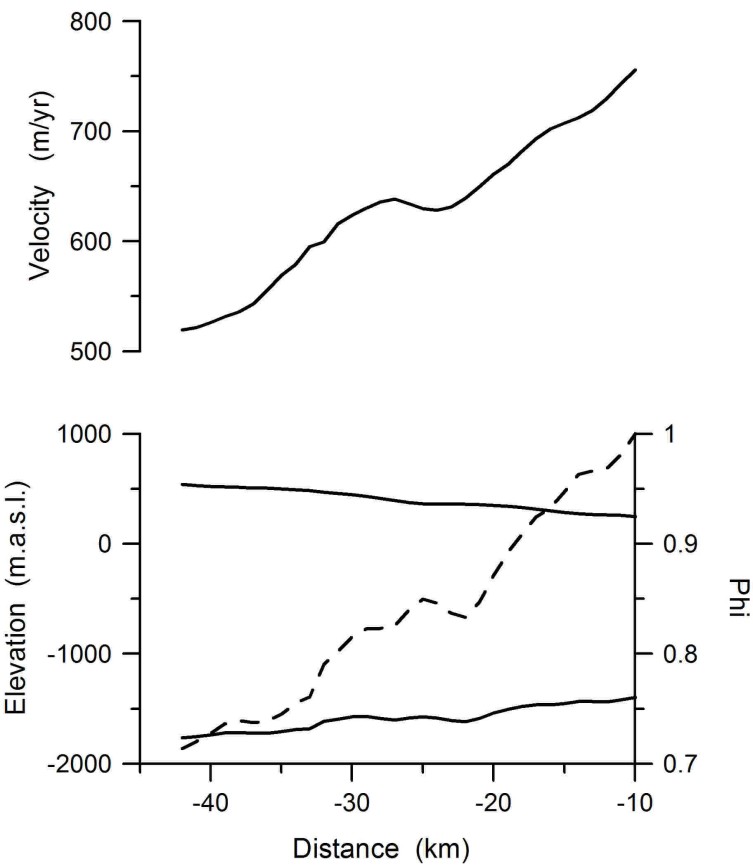

Figure 1.  Geometry of the lower part of Byrd Glacier, East Antarctica.  The dashed line in the lower panel
shows the buoyancy factor, calculated from eq. (12).





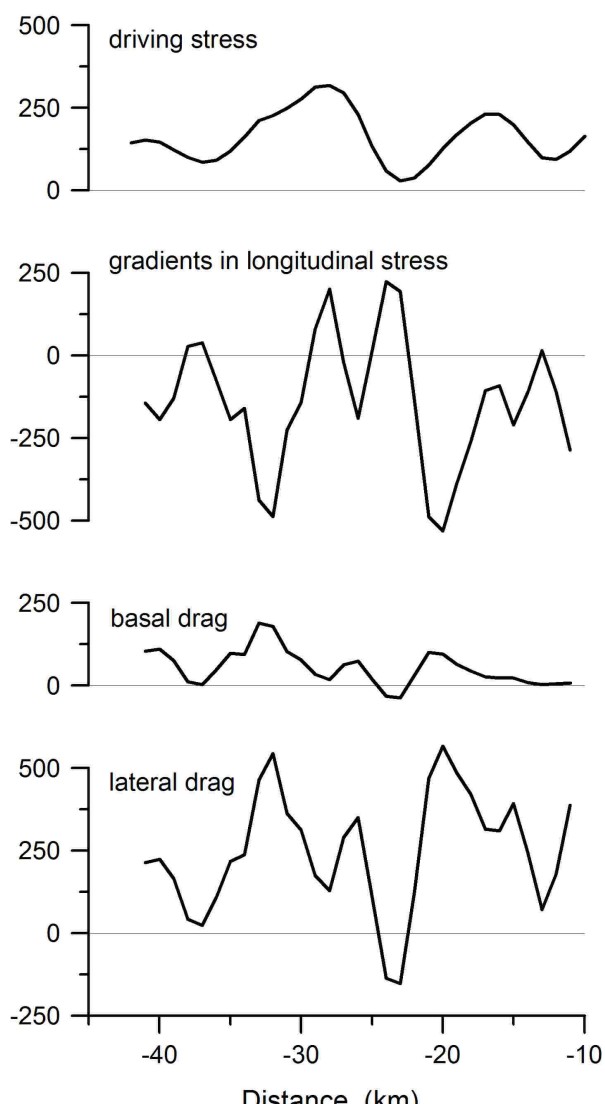

Figure 2. Force-balance terms according to geometric force balance, eqs. (13) – (14).



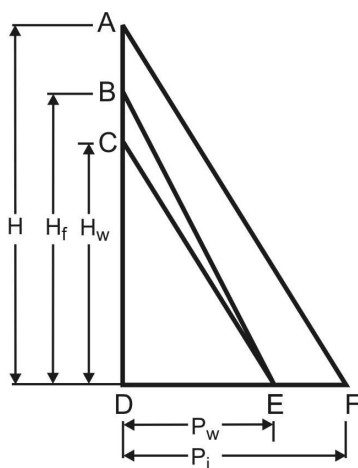

Figure 3.   Geometric force balance according to *Hughes* [2008].   H represents ice thickness, $H_f$ the
flotation height or height of the ice column supported by basal water pressure, and $H_w$ the piezometric
height; $P_w$ and $P_i$ represent the basal water pressure and weight of the ice column, respectively.   Ice flow is
from right to left.



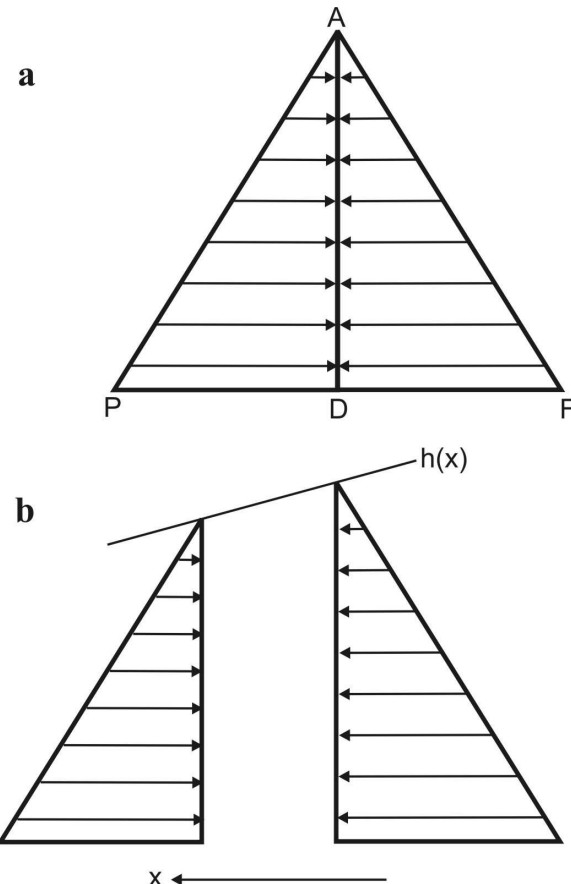

Figure 4.  (a) at any location the lithostatic stress increases linearly with depth from zero at the ice surface
to ρgH at the base; the lithostatic stress from ice on the right of the vertical line AD is balanced by an equal
but opposite lithostatic stress from ice on the right and the area of triangle ADF equals that of triangle
ADP.  (b) gradients in lithostatic stress are associated with a sloping ice surface, h(x), resulting in a smaller
lithostatic stress in the downslope direction; the difference between the areas of both triangles is a measure
of the gravitational driving stress responsible for glacier flow.