# Peer review of "Basal buoyancy and fast moving glaciers: in defense of analytic force balance"

_The Cryosphere, 2016_

## Referee Comment (RC1) · Anonymous Referee #1 · 8 Mar 2016

General comments:

Throughout the manuscript there is reference to a "Geometric Approach" of T. Hughes. It is not clear to me where this name comes from, as the most recent paper by Hughes (2016) that I reviewed for TCD does not have this in the title. I think that the present manuscript should very early on say specifically where this name appears in the long citation of Hughes' articles so that readers can go back and see where it first appears.

As far as I can tell from the TC article by Hughes (The Cryosphere, 10, 193-225, 2016), an application of the Geometric Approach is made to both Byrd and Jacobshavan Glaciers. The current manuscript reports to evaluate the implications Hughes approach by application to the Byrd Glacier (lines 38 and 39). I'm not sure, from looking at the 4 figures, where this evaluation is being made. Perhaps the brevity of lines

38 and 39 could be expanded to explain how this evaluation will be undertaken, i.e., will there be a direct comparison?

line 169-171, It's not clear to me why the result is "surprising" or that there is no credible mechanism. It's important to explain this, as many of the younger readers of the ice-stream literature may not have sufficient experience understanding how difficult early on it was to figure out how and why ice stream flow was possible.

A comment on lines 291 - 293: While this interpretation may have been true in 2009, exchanges of differing opinion and resolution of dispute in the North American nation where I live and work have led to deplorably mean rhetoric and disrespectful and sometime vulgar public discourse. I see this at times leaking over into the way in which colleagues (particularly in my nation) at times communicate to each other. I would thus say that the present manuscript under review does not rise to a state of sufficient negativity to be regarded as a personal attack anymore. In the past, it would be the role of the journal editors to make sure that any kind of vitriol is removed from a paper that comments on a colleague's work. In this day and age, particularly within the social milieu of where I live, it is hard to find language in manuscripts that would rise to the level of what has become commonplace.

Specific comments:

Line 337 - There is a typo in at least one reference. Given that citation accuracy has been criticized heavily in other journals serving the cryospheric community, I suggest that the Author provide proof that each and every citation has been proofread. Probably no such proof exists, however, it cannot be emphasized more that *someone* has to take responsibility for proof reading the citation list (and ensuring that it is accurate and in proper form).

---

## Referee Comment (RC2) · Anonymous Referee #2 · 26 Apr 2016

Overall, I find this manuscript to be clear and compelling. While I do not expect that the geometric force balance approach will supplant the analytical approach, this "defense" is likely to help readers to better understand the familiar equations. My comments thus do not contain any major objections, only requests for small clarifications and rewording of several phrases that appear (to me, at least) to contain sarcasm directed toward Hughes.

(33) While this analogy works for any specific type of sand, it is easy to imagine that one could "facilitate the forward motion" by pushing the cart onto more firmly packed sand, so the analogy does not consider variations in material properties of the bed. How (or whether) to modify this I leave to the author.

(45) Typo in equation, last term should have zz subscript.

[Figure]

(99) It would be nice to also present equation (7) with the definition of the driving stress (equation 3) substituted, in order to make the comparison between equations (7) and (9) immediate.

(117) For Section 3, at what point do you begin to disagree with Hughes? This could easily be interpreted as indicating that you agree with equations (10)-(12), because disagreement becomes clear at line 134.

(128) Typo: The inequality is written backwards.

(139) It may be more clear to also insert equation (6) here, because this is the form in which $F_S$ has appeared in most of the equations.

(142) While this is a good point, "The achievement here" is clearly sarcastic and should be rephrased.

(171) Does "This result" refer to the direction of the longitudinal stress gradient, the partitioning of the resistance, or both (in which case it should be plural)?

(187) In arguing against equation (16), it would be nice to point back to equation (3) for the driving stress for comparison, in addition to the reference to equation (7), after you have pretty much explained how to derive it.

(227) Based on what follows, it appears you are using equation (2) rather than (7).

(235) It would be helpful to include a short reminder that in introducing Glen's law you are substituting an expression in terms of deviatoric stresses into an equation in terms of full stresses. While you make this clear earlier, the reminder might help readers more quickly understand why $\sigma_{zz}$ appears in equation (19).

(248) "Thus, on a large scale...." doesn't include effects of basal buoyancy on $\tau_{bx}$, so it looks like you're contradicting yourself in the next paragraph.

(273) "The concept as presented by Hughes...." becomes unduly personal when the length of time and "has yet" are included.

(273) "phantom force" appears sarcastic; "phantom" is unnecessary in stating that a term is missing.

(290) The history of the manuscript is not really an acknowledgement, and mentioning that a past version was considered a personal attack on Hughes only suggests that this version is as well. I suggest cutting the second and third sentences.

---

## Author Comment (AC1) · 4 May 2016

Reply to Reviewer 1.

I am somewhat surprised by the general comment of Reviewer 1 regarding the use of the term "geometric force balance" especially because this reviewer acknowledges to have reviewed the most recent paper by Hughes (2016) for TCD. Admittedly, Hughes has never used the term in the title of his papers. Nevertheless, the term "geometric" originates with Hughes. Most recently, the 2016 paper in The Cryosphere (vol. 10, 1993-225) refers to "A geometrical force balance…." in the first sentence of section 3.1. According to the caption, Figure 5 shows "The geometrical force balance on an ice stream…." And on p. 201: "These are real stresses. They are obscured using holistic continuum mechanics in conventional ice-sheet models, but they visibly emerge from

the geometrical force balance in the holistic ice-sheet model based on Fig. 5." Similarly, the earlier paper submitted to The Cryosphere Discussions (vol. 8, 2043-2118, 2014) refers several times to "geometrical force balance." I did change "geometrical" to "geometric" to be in agreement with "analytic." An internet search failed to give an answer to which form is preferred or correct. My preference goes to "geometric" and "analytic."

The evaluation of the geometric approach to Byrd Glacier is presented in Section 4 and in Figures 1 and 2. I have added "on Byrd Glacier" to the caption of Figure 2.

Line 169-171: averaged over the length of the segment, gradients in longitudinal stress average -140 kPa, compared to a driving stress of 160 kPa. The minus sign indicates that the stress gradients work in the same direction as the driving stress: adding an additional force driving the glacier forward. I find this surprising. Longitudinal stress gradients may be important on a local scale of a few km, but cannot be as important as the driving stress over a distance of 30 km. The argument that "water buttressing produces a backstress" is not convincing, nor is the idea correct, as is explained in Section 5.

Line 291-293 has been deleted.

Line 337: my apologies for misspelling Whillans' name.
* * *

---

## Author Comment (AC2) · 4 May 2016

Reply to Reviewer 2.

Line 33: I will take the analogy out. Line 45: corrected. Line 99: done. Section 3: Interesting question. One can argue that eq. (10) represents the transition from inland-style flow to ice-shelf spreading. However, the way that phi is calculated in eq. (12) is incorrect, at least according to my interpretation. Line 128: changed. Line 139: done. Line 142: Leave out: "The achievement here is that" Line 173: The finding that longitudinal stress gradients act in cooperation with the driving stress over a distance of more than 30 km is surprising and there is no credible evidence that can explain this. Line 208: eq. (7) has been changed to include the expression (3) for the driving stress explicitly on the left-hand side. I added: with the term on the left-hand side describing

the driving stress. Line 227: yes, that should be equation (2) Line 235: added a sentence on why this term enters into eq. (17) Line 248: I don't understand the reviewer's comment. Line 273: Replace with: As shown in this contribution, the geometric force balance as presented by Hughes in a series of papers cannot be applied successfully to ice streams and outlet glaciers. Line 280: delete "phantom" Line 290: deleted